# Synthesis and Photocatalytic Activity of Novel Polycyclopentadithiophene

**DOI:** 10.3390/polym15204091

**Published:** 2023-10-15

**Authors:** Farah Zayanah Ahmad Zulkifli, Moeka Ito, Takahiro Uno, Masataka Kubo

**Affiliations:** Division of Applied Chemistry, Graduate School of Engineering, Mie University, 1577 Kurimamachiya-cho, Tsu 514-8507, Mie, Japan; kobunshi.gakusei@gmail.com (M.I.); uno@chem.mie-u.ac.jp (T.U.)

**Keywords:** photocatalyst, polycyclopentadithiophene, oxidative polymerization, oxidative hydroxylation

## Abstract

A novel π-conjugated polymer based on cyclopentadithiophene (CPDT) and poly(4,4′]-(((4*H*cyclopenta[2,1-*b*:3,4-*b*′]dithiophene-4,4-diyl)bis(ethane-2,1-diyl))bis(oxy))bis(4-oxobutanoic acid)) (PCPDT-CO_2_H) was prepared as a sparingly soluble material. The generation of hydroxyl radicals from PCPDT-CO_2_H in water was confirmed by using coumarin as a hydroxyl radical indicator. Furthermore, PCPDT-CO_2_H was found to catalyze the oxidative hydroxylation of arylboronic acid and the oxidation of benzaldehyde, indicating that PCPDT-CO_2_H can be a promising candidate for metal-free and 100% organic heterogeneous photocatalysts.

## 1. Introduction

Photocatalysts are highly effective materials that can quickly alter solar energy for various uses. The key characteristic of a photocatalyst is its ability to start and speed up chemical reactions by absorbing photons, without undergoing any long-term changes. For many applications, especially those involving the environment and energy, this characteristic is necessary. The most practical and commonly used photocatalyst is titanium dioxide (TiO_2_), due to its chemical stability, abundance, non-toxicity, and cost-effectiveness. TiO_2_ and other semiconductor photocatalysts have undergone substantial research for the oxidation of contaminants in water and air [1,2,3,4,5]. Other applications of photocatalysts include solar energy conversion [6,7,8,9,10,11] and organic synthesis [12,13,14,15,16,17,18]. However, TiO_2_ has the disadvantage of not being able to absorb visible light; it can only use ultraviolet (UV) light when exposed to sunlight. TiO_2_ has a relatively wide bandgap, typically around 3.2 to 3.4 electron volts (eV), which limits its electromagnetic spectrum absorption to the UV region [19,20]. This is a significant limitation of the material for practical use. TiO_2_ has a very low solar energy usage efficiency because the proportion of UV radiation to sunlight is only 3–5%. As a result, the research on photocatalysts is focused on the impregnation of TiO_2_ with the ability to respond to visible light and the development of methods that effectively convert visible light into usable energy [21,22,23,24]. Numerous modifications, such as ion doping, nano-structuring, and heterojunction construction, have been researched to enhance the photocatalytic capabilities of TiO_2_ [25,26].

Currently, π-conjugated materials attract considerable scientific interest as organic heterogeneous photocatalysts. This is primarily due to their distinctive characteristics and remarkable capacity to effectively capture visible light. The effective visible light absorption is contributed by the extended electron conjugation along the molecular structures of π-conjugated materials. Another attractive characteristic of π-conjugated materials is their tunable properties. By altering the chemical structures of π -conjugated materials, researchers may precisely control their tunable properties. This tunability allows optimization of their photocatalytic activity for various applications. Representative examples of π-conjugated materials include graphitic carbon nitride (g-C_3_N_4_) [27,28,29] and conjugated microporous polymers (CMPs) [30,31,32]. These materials are new classes of metal-free catalysts based on earth-abundant carbon materials. A study by Han et al. explored the selection of an electron donor for CMP, to construct a high-performance polymer catalyst. The researchers discovered that the photocatalytic activity of the resulting polymers was greatly influenced by the geometry of the electron donor. An efficient electron donor for CMP promotes a better charges transmission and light-induced electron/hole separation [33]. CMPs are characterized by a highly conjugated structure, where alternating single and multiple bonds create delocalized π-electron systems. Their versatility, tunability, and ability to generate reactive intermediates make them valuable tools in the field of photocatalysis, contributing to advancements in renewable energy, environmental sustainability, and green chemistry practices [30,31,32,33].

As mentioned above, photocatalysts, starting with titanium dioxide, are being developed into materials with higher performance and environmental compatibility, such as visible light-responsive and metal-free types. The design of photocatalysts for wide-range light collection, from ultraviolet to NIR regions, is an efficient way to accomplish the practical use of photocatalysis, since solar light contains around 50% near-infrared (NIR) light. The full utilization of sunlight is valuable in terms of efficient use of energy, and, furthermore, near-infrared rays are highly permeable, making them convenient for use as an environmental remediation material. The study of NIR-responsive materials has so far focused on sensitization using NIR-responsive substances including dye molecules and black phosphorus, the surface plasmon resonance effect, up-conversion, and materials with narrow band gaps that act as NIR harvesters [34,35,36].

We were interested in a 4*H*-cyclopenta[2,1-*b*:3,4-*b*′]dithiophene (CPDT)-based self-doped intrinsically conducting polymer as a new candidate for NIR-harvesting material, because Zotti et al. carried out anodic coupling polymerization of CPDT-based monomers and found that the resulting polymers showed a large absorption at the NIR light region [37,38]. Recently, we synthesized a novel self-doped conducting polymer based on CPDT and poly(4,4′-(((4*H*-cyclopenta[2,1-*b*:3,4-*b*′]dithiophene-4,4-diyl)bis(ethane-2,1-diyl))bis(oxy)) bis(butane-1-sulfonic acid)) (PCPDT-SO_3_H) [39]. It was found that PCPDT-SO_3_H exhibited a large absorption above 800 nm, which is characteristic for the oxidized conducting state of polythiophenes [36,38,40]. Although PCPDT-SO_3_H may be a potent candidate as a photocatalyst for NIR harvesting, it is difficult to use PCPDT-SO_3_H as a heterogeneous photocatalyst due to its high solubility in water. Our idea to decrease this solubility is to introduce carboxyl functionalities onto the side chain of the π-conjugated main chain. McCullough et al. reported that poly(3-(2-carboxyethyl)thiophene) exhibits low solubility in conventional organic solvents [41]. We will report here the preparation of poly(4,4′-(((4*H*-cyclopenta[2,1-*b*:3,4-*b*′]dithiophene-4,4-diyl)bis(ethane-2,1-diyl))bis(oxy))bis(4-oxobutanoic acid)) (PCPDT-CO_2_H, Figure 1) as a sparingly soluble material. Further, the photocatalytic activities of PCPDT-CO_2_H will be examined.

## 2. Materials and Methods

### 2.1. Materials

3-Bromothiophene (**1**), thiophene-3-carboxaldehyde (**2**), 2-(2-Bromoethoxy)tetrahydro-2*H*-pyran, pyridinium chlorochromate (PCC), iodine, and pyridinium *p*-toluenesulfonate (PPTS) were purchased from Tokyo Chemical Industry. Butyllithium in hexane (1.6 M), copper powder (particle size: 75 μm–150 μm), and anhydrous iron (III) chloride were purchased from Kanto Chemical Co., Inc. (Tokyo, Japan). All other reagents were obtained from commercial sources and used as received.

### 2.2. Compounds

#### 2.2.1. Bis(2-iode-3-thienyl)methanol (**3**)

3-Bromothiophene (**1**) (6.52 g, 40 mmol) was dissolved in 50 mL of ether and the solution was cooled to −78 °C. Then, 25 mL of *n*-butyllithium in hexane (1.6 M) was added via a syringe, and the mixture was stirred under nitrogen for 3 h at −78 °C. A solution of thiophene-3-carbaldehyde (**2**) (4.48 g, 40.0 mmol) in ether (40 mL) was added, and the mixture was allowed to warm to room temperature. Then, the reaction mixture was cooled to −20 °C and 50 mL of *n*-butyllithium in hexane (1.6 M) was added via a syringe. The reaction mixture was stirred at −20 °C for 2 h and allowed to warm to room temperature. After cooling to −20 °C, a solution of ether (200 mL) containing 32.0 g (126 mmol) of iodine was added dropwise. The reaction mixture was allowed to warm to room temperature and was left overnight. Following the completion of the reaction, the reaction mixture was washed with an aqueous Na_2_S_2_O_3_ solution and water. The organic layer was dried over anhydrous magnesium sulfate and placed under reduced pressure to remove the solvent. The resulting brown residue was triturated with carbon tetrachloride, filtered, and further washed with carbon tetrachloride to obtain 9.6 g (54%) of compound **3** as a beige powder; ^1^H NMR (500 MHz, CDCl_3_, δ): 7.43 (d, *J* = 6.0 Hz, 2H), 6.97 (d, *J* = 6.0 Hz, 2H), 5.76 (s, 1H), 2.26 (s, 1H); ^13^C NMR (125 MHz, CDCl_3_, δ): 146.8, 131.5, 127.0, 76.84, 71.84; IR (NaCl, cm^−1^): 3349, 3098, 698, 49.

#### 2.2.2. Bis(2-iodo-3-thienyl)ketone (**4**)

To a stirred solution of compound **3** (9.6 g, 21 mmol) in 200 mL of dichloromethane was added pyridinium chlorochromate (PCC) (6.9 g, 32 mmol), and the mixture was stirred at room temperature for 12 h. After the reaction completed, the resulting slurry was passed through a silica gel column, using dichloromethane as the eluent, to obtain 9.5 g (quant) of compound **4** as a brown solid; ^1^H NMR (500 MHz, CDCl_3_, δ): 7.46 (d, *J* = 5.5 Hz, 2H), 7.04 (d, *J* = 6.0 Hz, 2H); ^13^C NMR (125 MHz, CDCl_3_, δ): 185.7, 143.3, 131.7, 129.9, 81.4; IR (NaCl, cm^−1^): 3102, 1652, 728, 699.

#### 2.2.3. 4*H*-Cyclopenta[2,1-*b*:3,4-*b*′]dithiophen-4-one (**5**)

To a solution of compound **4** (9.4 g, 21 mmol) in 60 mL of *N*,*N*-dimethylformamide (DMF) was added Cupper (4.0 g, 63 mmol), and the reaction mixture was heated under reflux for 4 h. After the reaction completed, the solid was removed by filtration through celite. Ether was added to the filtrate and washed with water three times. The organic layer was dried over anhydrous magnesium sulfate and placed under reduced pressure to remove the solvent. The residue was purified by passing through a silica gel column, using dichloromethane as the eluent, to obtain 3.5 g (87%) of compound **5** as a reddish purple solid; ^1^H NMR (500 MHz, CDCl_3_, δ): 7.03 (d, *J* = 5.5 Hz, 2H), 6.98 (d, *J* = 6.0 Hz, 2H); ^13^C NMR (125 MHz, CDCl_3_, δ): 182.8, 149.3, 142.5, 127.2, 121.8; IR (NaCl, cm^−1^): 3104–3074, 1699, 691.

#### 2.2.4. 4*H*-Cyclopenta[2,1-*b*:3,4-*b*′]dithiophene (CPDT)

A mixture of compound **5** (3.5 g, 18 mmol), ethylene glycol (60 mL), hydrazine hydrate (14 mL), and finely grounded potassium hydroxide (3.5 g, 52 mmol) was heated at 180 °C under nitrogen atmosphere for 24 h. After the completion of the reaction, the reaction mixture was allowed to cool to room temperature. Then, the reaction mixture was poured into water and extracted with dichloromethane three times. The combined organic layer was washed with water, brine, and saturated ammonium chloride solution. The organic layer was dried over anhydrous magnesium sulfate and placed under reduced pressure to remove the solvent. The resulting brown oil was passed through a silica gel column, using hexane as the eluent, to obtain 2.4 g (72%) of CPDT as a colorless solid; ^1^H NMR (500 MHz, CDCl_3_, δ): 7.18 (d, *J* = 5.8 Hz, 2H), 7.08 (d, *J* = 5.8 Hz, 2H), 3.53 (s, 2H); ^13^C NMR (125 MHz, CDCl_3_, δ): 149.7, 138.5, 124.5, 123.0, 31.9; IR (NaCl, cm^−1^): 3089, 687.

#### 2.2.5. 2,2′-(((4*H*-Cyclopenta[2,1-*b*:3,4-*b*′]dithiophene-4,4-diyl)bis(ethane-2,1-diyl))bis(oxy)) bis(tetrahydro-2H-pyran) (CPDT-OTHP)

Finely crushed potassium hydroxide (1.0 g) was added to a solution containing CPDT (1.0 g, 5.6 mmol), 2-(2-bromoethoxy)tetrahydro-2H-pyran (2.4 g, 11.2 mmol), and potassium iodide (20 mg) in dimethyl sulfoxide (DMSO) (20 mL). The resulting mixture was stirred for 24 h at room temperature. Dichloromethane was used to extract the reaction mixture after it had been placed into water. After being dried over anhydrous magnesium sulfate, the organic layer was washed with water and placed under reduced pressure to remove the solvent. A silica gel column was used to charge the residue, and the eluent used was a mixture of hexane and ethyl acetate (4:1 *v*/*v*). In the first band, 2.2 g (90%) of CPDT-OTHP was obtained as a light yellow oil; ^1^H NMR (CDCl_3_, δ): 7.16 (d, *J* = 5.8 Hz, 2H), 6.98 (d, *J* = 5.8 Hz, 2H), 4.26 (t, *J* = 5.9 Hz, 2H), 3.7–2.9 (m, 8H), 2.25 (d, *J* = 7.3 Hz, 4H), 1.8–1.3 (m, 12H); ^13^C NMR (CDCl_3_, δ): 156.5, 136.6, 124.8, 121.7, 98.8, 63.9, 61.6, 49.3, 37.7, 30.4, 25.2, 19.2; IR (NaCl, cm^−1^): 2939, 1120, 1076, 1131, 669; Anal. calcd. for C_23_H_30_O_4_S_2_: C 63.56, H 6.96; found: C 63.39, H 7.06.

#### 2.2.6. 2,2′-(4*H*-Cyclopenta[2,1-*b*:3,4-*b*′]dithiophene-4,4-diyl)bis(ethan-1-ol) (CPDT-OH)

PPTS (0.4 g, 1.6 mmol) was added to a solution of CPDT-OTHP (2.0 g, 4.6 mmol) in 40 mL of ethanol, and the mixture was stirred for 5 h at 60 °C. Diethyl ether was used to extract the reaction mixture after it had been placed into water. After being dried over anhydrous magnesium sulfate, the organic layer was washed with water and placed under reduced pressure to remove the solvent. Ethyl acetate was used as the eluent and the residue was charged onto a silica gel column. A white solid weighing 1.0 g (79%) of CPDT-OH was obtained from the first band; ^1^H NMR (500 MHz, CDCl_3_, δ): 7.32 (d, *J* = 5.2 Hz, 2H), 7.07 (d, *J* = 5.2 Hz, 2H), 3.12 (t, *J* = 6.8 Hz, 4H), 2.24 (t, *J* = 6.8 Hz, 4H); ^13^C NMR (125 MHz, CDCl_3_, δ): 158.0, 137.6, 126.5, 122.6, 59.3, 50.5, 41.4; IR (KBr, cm^−1^): 3273, 2902, 1451, 1043, 676; Anal. calcd. for C_13_H_14_O_2_S_2_: C 58.62, H 5.30; found: C 58.79, H 5.16.

#### 2.2.7. 4,4′-(((4*H*-Cyclopenta[2,1-*b*:3,4-*b*′]dithiophene-4,4-diyl)bis(ethane-2,1-diyl))bis(oxy)) bis(4-oxobutanoic acid) (CPDT-CO_2_H)

A mixture of 2,2′-(4*H*-cyclopenta[2,1-*b*:3,4-*b*′]dithiophene-4,4-diyl)bis(ethan-1-ol) (CPDT-OH) (0.25 g, 0.94 mmol), succinic anhydride (0.19 g, 1.9 mmol), 4-dimethylaminopyridine (DMAP) (0.23 g, 1.9 mmol), and tetrahydrofuran (THF) (20 mL) was heated under reflux for 48 h. The reaction mixture was added to diluted hydrochloric acid and extracted with ethyl acetate. Anhydrous magnesium sulfate was used to dry the organic layer, and the solvent was removed by placing it under reduced pressure. Ethyl acetate was used as the eluent while charging the residue onto a silica gel column. The first band was collected to give 0.44 g (quant) of CPDT-CO_2_H as a viscous green oil; ^1^H NMR (500 MHz, CDCl_3_, δ): 7.19 (d, *J* = 5.5 Hz, 2H), 6.99 (d, *J* = 5.5 Hz, 2H), 3.75 (t, *J* = 6.8 Hz, 4H), 2.58 (m, 4H), 2.46 (m, 4H), 2.25 (t, *J* = 6.8 Hz, 4H); ^13^C NMR (125 MHz, CDCl_3_, δ): 177.7, 177.1, 155.3, 137.3, 125.8, 121.4, 61.7, 49.3, 36.4, 28.9, 28.8; IR (NaCl, cm^−1^): 2930, 1716; Anal. calcd. for C_21_H_22_O_8_S_2_: C 54.07, H 4.75; found: C 54.28, H 4.91.

#### 2.2.8. Poly(4,4′-(((4*H*-cyclopenta[2,1-*b*:3,4-*b*′]dithiophene-4,4-diyl)bis(ethane-2,1-diyl))bis (oxy))bis(4-oxobutanoic acid)) (PCPDT-CO_2_H)

A mixture of CPDT-CO_2_H (0.61 g, 1.3 mmol), FeCl_3_ (5.2 mmol), and chloroform (20 mL) was stirred under nitrogen for 24 h. The reaction mixture was poured into methanol (120 mL) containing a few drops of hydrazine monohydrate. The precipitated polymer was purified by Soxhlet extraction, using methanol as a solvent, to obtain PCPDT-CO_2_H (0.43 g, 70%) as a black solid with a metallic luster.

### 2.3. Hydroxy Radical Detection

A 3 mL of 0.4 mM aqueous coumarin solution was added to PCPDT-CO_2_H (15 mg), and the mixture was irradiated by a Xe lamp (100 W) at room temperature for 24 h. Following the process of filtration, the resulting filtrate was subjected to analysis using a fluorescence spectrometer.

### 2.4. Photocatalytic Oxidative Hydroxylation of 1,4-Phenylenediboronic Acid

Triethylamine (61 mg, 0.6 mmol), *N*,*N*-dimethylformamide (DMF), PCPDT-CO_2_H (15 mg), 1,4-phenylenediboronic acid (33 mg, 0.2 mmol), and DMF (1 mL) were mixed and exposed to a Xe lamp (100 W) at room temperature for 3 h. After adding the reaction mixture to 10% hydrochloric acid, ether was used to extract it. The organic layer was dried with anhydrous magnesium sulfate and placed under reduced pressure to remove the solvent. The ^1^H NMR was used to analyze the residue.

### 2.5. Photocatalytic Oxidation of Benzaldehyde

A mixture of PCPDT-CO_2_H (6 mg), benzaldehyde (45 mg), and chloroform (1.5 mL) was irradiated by a Xe lamp (100 W) at room temperature for 3 h. The solid catalyst was removed from the mixture by filtering, and the solvent was eliminated by applying lower pressure to the filtrate. The residue was analyzed using ^1^H NMR.

### 2.6. Measurements

Nuclear magnetic resonance spectra (NMR) were recorded at 500 MHz for ^1^H spectra and 125 MHz for ^13^C spectra (ECZ500R, JEOL, Akishima, Japan). The analysis was conducted at room temperature. The samples were dissolved in CDCl_3_, with tetramethylsilane (TMS) serving as the internal standard. Photoluminescence spectra were recorded on a HAMAMATSU Multi Channel Analyzer PMA-11 (Hamamatsu, Japan). The analysis was conducted at the exciting wavelength of 365 nm. Fourier-transform infrared (FTIR) spectra were were recorded on a JASCO FT/IR-4100 (Tokyo, Japan). UV-vis-NIR spectra were recorded on a JASCO V-770 (Tokyo Japan). Elemental analysis was carried out using YANACO CHN-corder MT-5 (Kyoto, Japan).

## 3. Results and Discussion

### 3.1. Preparation of PCPDT-CO_2_H

Figure 2 shows the synthetic pathway of PCPDT-CO_2_H. The key compound is CPDT. Several synthetic pathways have been reported for the synthesis of CPDT [42,43,44]. Based on the results of our preliminary experiments, we synthesized CPDT by modifying the procedure [45] reported by Pal et al. CPDT was synthesized using 3-bromothiophene (**1**) and thiophene-3-carbaldehyde (**2**) as starting materials in 4 steps. In order to introduce carboxyl functionalities on the side chain of CPDT, we first carried out condensation reaction between CPDT and 2-(2-bromoethoxy)tetrahydro-2*H*-pyran, to obtain CPDT-OTHP that carries hydroxyl functionality masked as the 2-tetrahydopyranyl ether (THP group). This is because the THP group is stable during condensation reactions under alkaline conditions. Then, we carried out the deprotection of THP group to liberate the hydroxyl functionalities. Finally, the reaction of CPDT-OH with succinic anhydride gave the desired monomer, CPDT-CO_2_H, utilizing a well-known reaction between alcohol and succinic anhydride. The structure of CPDT-CO_2_H was confirmed by ^1^H and ^13^C NMR spectra, as shown in Figure 3 and Figure 4, respectively. These Figures show peak assignments.

The obtained CPDT-CO_2_H was polymerized in chloroform by oxidative polymerization, using FeCl_3_ as an oxidant. It is a common strategy for synthesizing certain conjugated polymers through oxidative polymerization [46,47]. This approach is often referred to as chemical oxidative polymerization or oxidative coupling. Oxidative polymerization is a well-established method for the synthesis of conjugated polymers with π-conjugated backbones. It allows for the efficient coupling of monomers to form extended conjugated systems. This method can lead to the formation of high-molecular-weight polymers with desirable properties for applications in organic electronics, photovoltaics, and other fields. The choice of chloroform and FeCl_3_ in oxidative polymerization offers good control over the reaction conditions and polymerization process, leading to consistent and reproducible results. The resulting polymer, PCPDT-CO_2_H, was purified by Soxhlet extraction using methanol as the solvent. PCPDT-CO_2_H was obtained as a black solid with a metallic luster, a typical physical appearance of self-doped conducting polymers. Figure 5 shows the IR spectrum of PCPDT-CO_2_H. The broad OH-band in the region from 3700 to 3100 cm^−1^ is due to carboxyl functionalities. The C-H stretching vibrations of methylene moiety are observed around 2920 cm^−1^. The peaks at 1720 and 1630 cm^−1^ are due to the C=O stretching of ester and carboxylate, respectively. PCPDT-CO_2_H was partially soluble in THF, but insoluble in hexane, chloroform, acetonitrile, methanol, *N*,*N*-dimethylformamide (DMF), and water. Therefore, PCPDT-CO_2_H is a suitable material for heterogeneous photocatalysts for organic synthesis. Figure 6 shows the UV–visible–NIR spectrum of PCPDT-CO_2_H in the solid state. The sharp absorption at around 350 nm is probably due to the source changeover between the tungsten lamp (visible region) and the deuterium discharge lamp (ultraviolet region). The spectrum exhibited a broad continuous absorption from 400 to 800 nm. The optical bandgap was determined to be 1.7 eV. Unlike PCPDT-SO_3_H, PCDT-CO_2_H did not exhibit a large absorption in NIR region. This is probably due to the weaker acidity of CO_2_H than SO_3_H. Therefore, we examined photocatalytic activities of CPDT-CO_2_H under visible light irradiation conditions.

### 3.2. Generation of Hydroxyl Radical

The detection of hydroxyl radicals was carried out to better understand and keep track of chemical processes. Coumarin is a commonly used probe for the detection of hydroxyl radicals in heterogeneous photocatalytic reactions [48]. The use of coumarin is based on its ability to undergo a reaction with hydroxyl radicals, leading to the formation of a fluorescent product. This fluorescence change can be monitored spectroscopically, which allows for the study of the presence and activity of hydroxyl radicals in photocatalytic systems. The hydroxy radical reacts with coumarin to give umbelliferone, which is used to detect hydroxyl radical by fluorometry. In this work, 0.4 mM coumarin aqueous solution was photo-irradiated in the presence of PCPDT-CO_2_H. The reaction mixture was filtered to remove PCPDT-CO_2_H, and the filtrate was investigated by PL spectrum. Figure 7 shows the PL spectra of the coumarin solution before and after photoirradiation. A sharp emission at 365 nm is from UV-LED as an excitation light source. The observed spectra were normalized using this emission at 365 nm. After irradiation, emission at 450 nm was observed, and this emission is coming from umbelliferone, suggesting that hydroxyl radicals were generated when visible light was irradiated. PCPDT-CO_2_H served as the photocatalyst in the system. PCPDT-CO_2_H absorbs photons from the visible light, creating electron–hole pairs within the material. This absorption of photons is what allows the photocatalyst to become activated. Once activated, PCPDT-CO_2_H can participate in redox reactions. It can transfer electrons from the excited state to nearby molecules or species to generate hydroxyl radicals. The hydroxyl radicals generated by PCPDT-CO_2_H are then available to react with coumarin and produce umbelliferone. This finding shows that PCPDT-CO_2_H can effectively serve as a photocatalyst for harnessing the energy of visible photons to drive chemical reactions.

### 3.3. Oxidative Hydroxylation of 1,4-Phenylenediboronic Acid

Luo et al. [49] and Wang et al. [50] reported that conjugated microporous polymer (CMP) could serve as a reusable and efficient visible light heterogeneous photocatalyst for the oxidative hydroxylation of arylboronic acids, using molecular oxygen as a green oxidant. Therefore, we examined the photocatalytic activity of PCPDT-CO_2_H in the photocatalytic oxidative hydroxylation of arylboronic acid under visible light irradiation. The photocatalytic oxidative hydroxylation of 1,4-phenylenediboronic acid (**1**) was chosen as the model reaction. The boronic acid (**1**) undergoes a hydroxylation reaction to give 4-hydroxyphenylboronic acid (**2**), which is finally converted to hydroquinone (**3**). An example of a ^1^H NMR spectrum of the reaction products is shown in Figure 8 with peak assignments. The reaction mixtures were composed of hydroquinone (**3**), 4-hydroxyphenylboronic acid (**2**), and unreacted 1,4-pheneylenediboronic acid (**1**). The percent composition of the reaction products was determined from the integral ratio of peaks a, e and g. We carried out a series of screening and control experiments. Table 1 summarizes the experimental results. It is obvious that CPDT-CO_2_H, triethylamine, oxygen, and light are essential in this oxidative hydroxylation reaction. These experimental results suggest a reaction mechanism similar to the literature [49], as shown in Figure 9. The charge separation occurs when PCPDT-CO_2_H is irradiated by light, as some of its electrons are promoted to higher energy levels. This creates electron–hole pairs in the material. Due to the presence of electron-rich and electron-deficient regions in the PCPDT-CO_2_H material, the excited electrons can become separated from the holes (electron vacancies). In the presence of oxygen (O_2_), the excited electrons can reduce molecular oxygen to form superoxide anions (O_2_^•−^). This process occurs because the excited electrons have enough energy to reduce molecular oxygen to superoxide. The generated superoxide anions (O_2_^•−^) can act as reactive intermediates and participate in chemical reactions by reacting with boronic acid. On the other hand, triethylamine acts as a reducing agent and converts the boronic acid moiety to a hydroxyl group. The overall sequence of events involves light absorption, charge separation, superoxide anion generation, and a subsequent reaction with the boronic acid. This process is typical in photocatalytic reactions, where light energy is harnessed to drive chemical transformations.

### 3.4. Oxidation of Benzaldehyde

Finally, we demonstrated that PCPDT-CO_2_H efficiently could catalyze the oxidation of benzaldehyde under photoirradiation. PCPDT-CO_2_H was added to a chloroform solution of benzaldehyde. Upon photoirradiation (using a Xe lamp) of the reaction mixture, the conversion from benzaldehyde to benzoic acid was estimated using ^1^H NMR (Figure 10). After 3 h of irradiation, the conversion reached 70%. Since benzaldehyde is easily oxidized in the air [51,52], control experiments were performed. Under the same irradiation conditions without PCPDT-CO_2_H, the conversion was only 10%. No reaction occurred when PCPDT-CO_2_H was added to the benzaldehyde solution in the dark for 3 h. These experimental results indicated that PCPDT-CO_2_H efficiently catalyzed the oxidation of benzaldehyde under photoirradiation.

## 4. Conclusions

The FeCl_3_-based oxidative polymerization of CPDT-CO_2_H was carried out to obtain PCPDT-CO_2_H as a new photocatalyst. It shows a strong and broad absorption in the visible light region. After photo irradiation of the dispersion of PCPDT-CO_2_H in aqueous coumarin solution, we observed a blue emission from the reaction mixtures. The emission comes from umbelliferone, indicating that hydroxyl radicals were generated during photoirradiation. Further, PCPDT-CO_2_H was found to catalyze oxidative hydroxylation of arylboronic acid. Finally, photoirradiation of chloroform solution of benzaldehyde was carried out in the presence of PCPDT-CO_2_H, to find that benzaldehyde was oxidized to benzoic acid efficiently. These experiment results indicate that PCPDT-CO_2_H is a promising candidate for a metal-free and 100% organic photocatalyst.

## Figures and Tables

**Figure 1 polymers-15-04091-f001:**
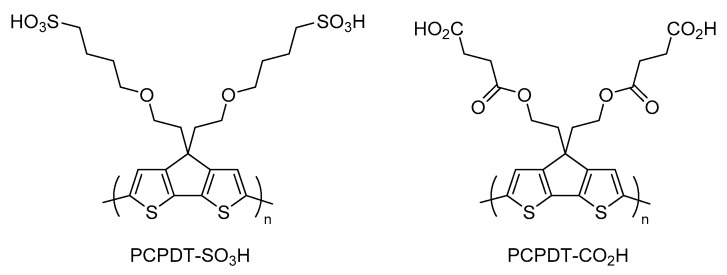
Chemical structure of PCPDT-SO_3_H and PCPDT-CO_2_H.

**Figure 2 polymers-15-04091-f002:**
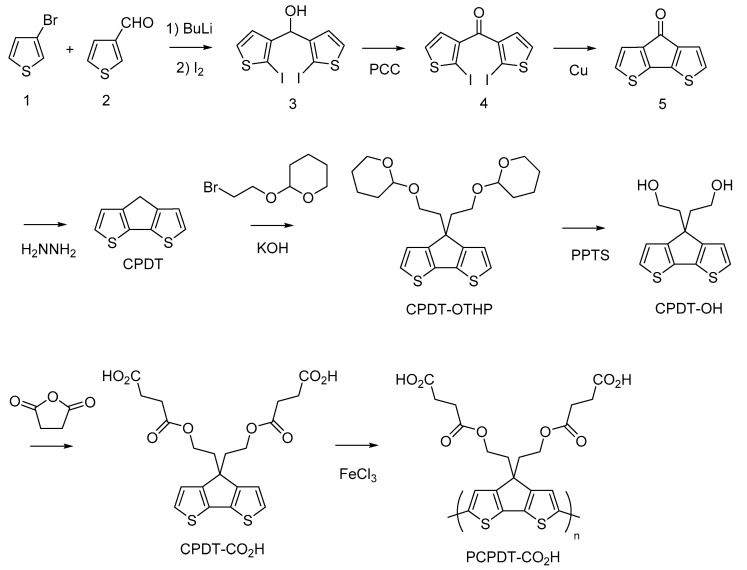
Synthetic pathway of PCPDT-CO_2_H.

**Figure 3 polymers-15-04091-f003:**
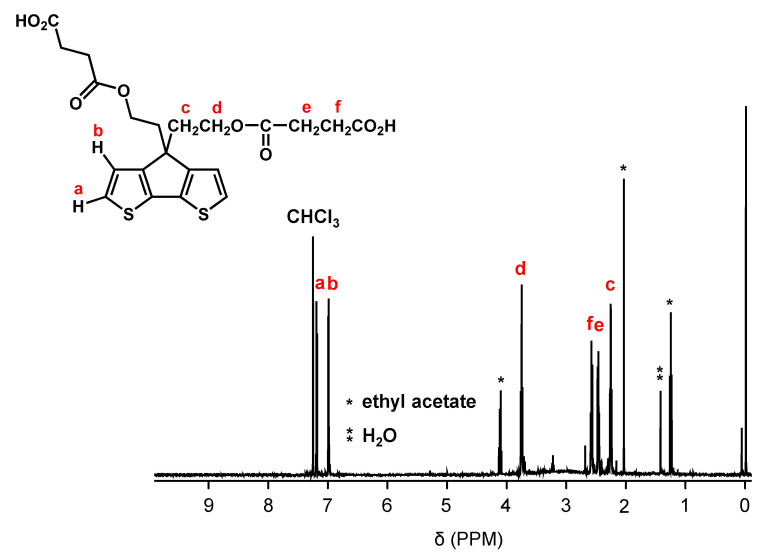
^1^H NMR spectrum of CPDT-CO_2_H in CDCl_3_.

**Figure 4 polymers-15-04091-f004:**
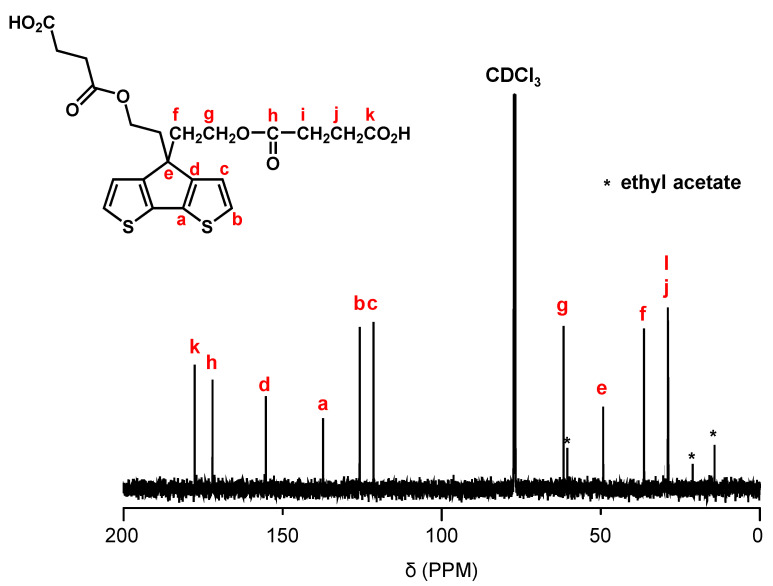
^13^C NMR spectrum of CPDT-CO_2_H in CDCl_3_.

**Figure 5 polymers-15-04091-f005:**
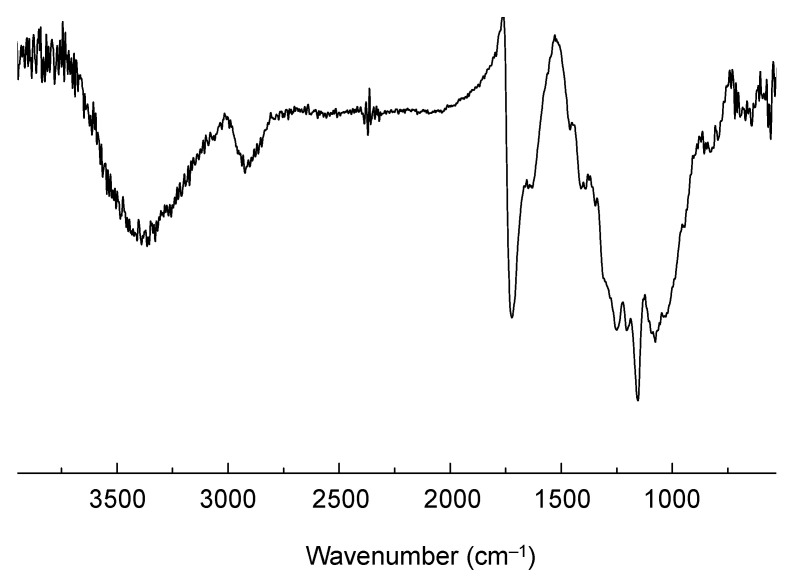
IR spectrum of PCPDT-CO_2_H.

**Figure 6 polymers-15-04091-f006:**
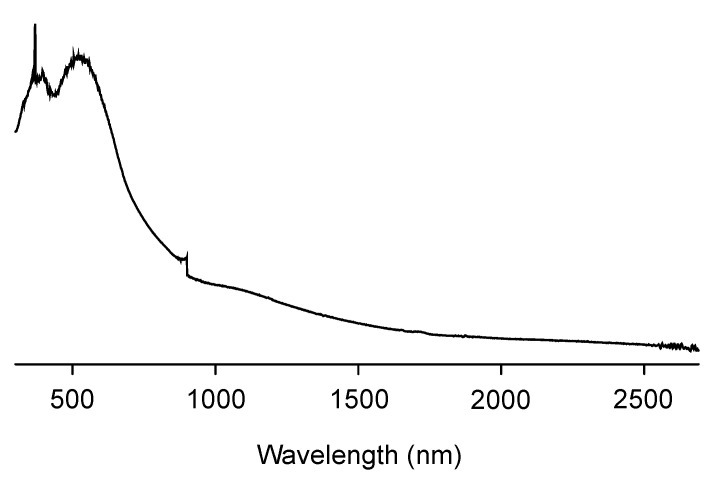
Solid-state UV–visible–NIR absorption spectrum of PCPDT-CO_2_H.

**Figure 7 polymers-15-04091-f007:**
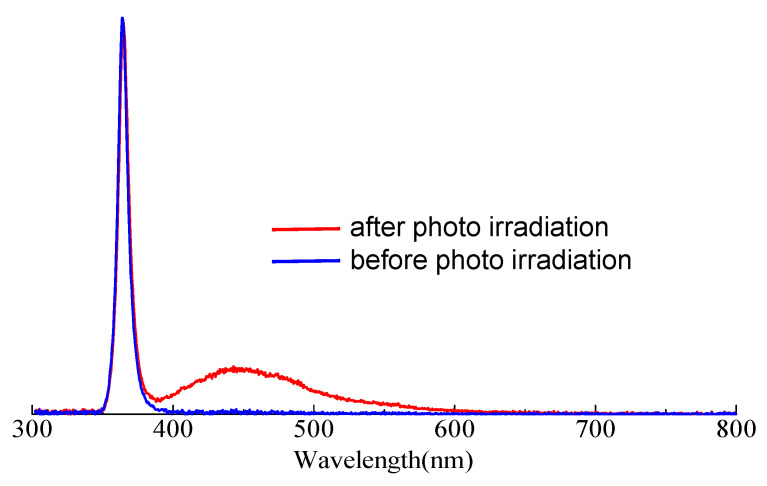
PL spectra of aqueous coumarin solution, (blue line) before and (red line) after photo irradiation (λ_ex_ = 365 nm).

**Figure 8 polymers-15-04091-f008:**
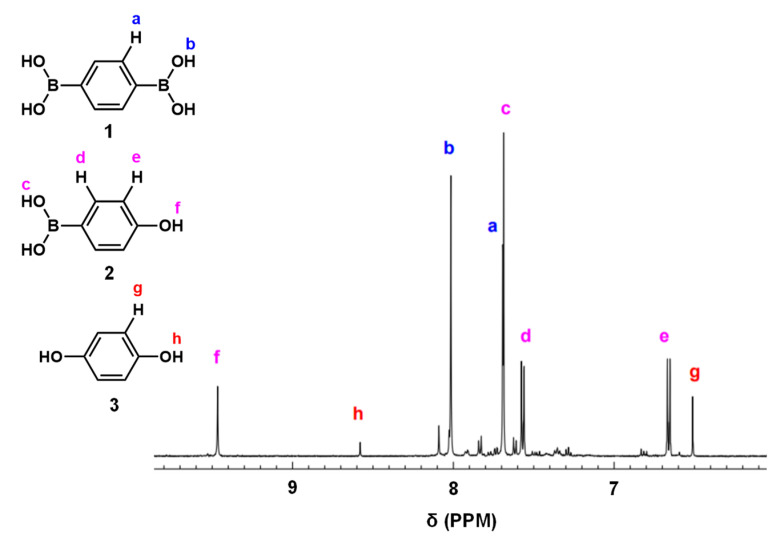
An example of ^1^H NMR spectrum of the reaction products (entry 1, Table 1).

**Figure 9 polymers-15-04091-f009:**
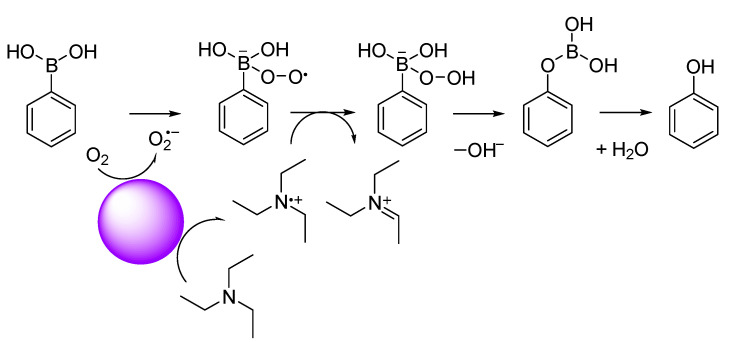
Plausible reaction mechanism for hydroxylation of phenylboronic acid.

**Figure 10 polymers-15-04091-f010:**
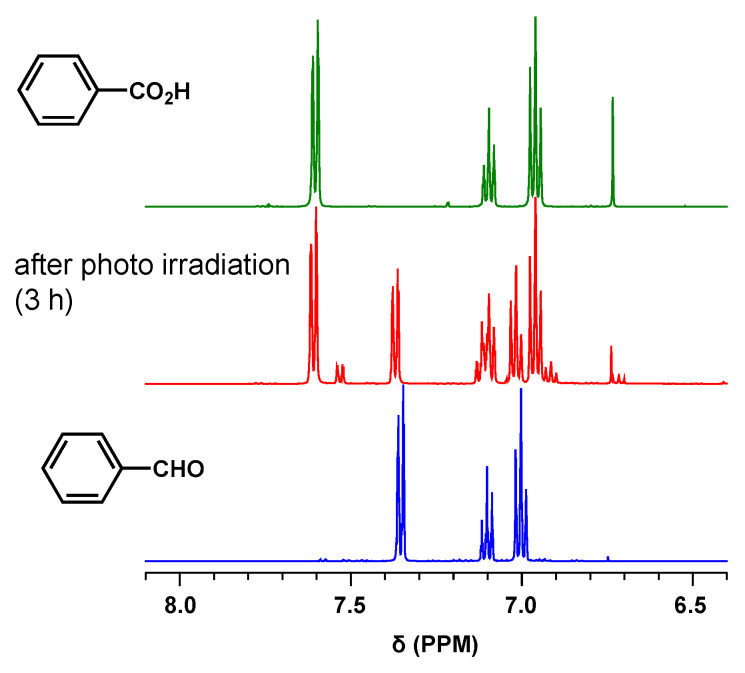
^1^H NMR spectra of benzoic acid, benzaldehyde, and reaction mixture in CDCl_3_.

**Table 1 polymers-15-04091-t001:** Results of oxidative hydroxylation of 1,4-Phenylenediboronic acid.

		Composition (mol%)		
Entry	Reaction Conditions	1	2	3
1	normal ^1^	32	58	10
2	w/o PCPDT-CO_2_H	100	0	0
3	in the dark	100	0	0
4	w/o Et_3_N	100	0	0
5	w/o O_2_ ^2^	95	4	1
6	w/radical scavenger ^3^	97	2	1

^1^ **1** = 0.2 mmol; Et_3_N = 0.6 mmol, PCPDT-CO_2_H = 15 mg: DMF = 1 mL; Xe lamp (100 W) = 3 h. ^2^ N_2_ bubbling. ^3^ Benzoquinone = 5 mg.

## Data Availability

The data presented in this study are available on request from the corresponding author.

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
