# Peer review of "Synthesis and Photocatalytic Activity of Novel Polycyclopentadithiophene"

_polymers, 2023, doi:10.3390/polym15204091_

Round 1
Reviewer 1 Report
This paper is interesting in the field of metal-free organic photocatalysts.
I recommend this paper for Polymers with very minor revisions as follows.
If possible, the rough estimation of the quantum efficiency of the photocatalyst will be mentioned.
The misspelling is in line 79 and 115: metaric lustra → metaric luster
Author Response
Thank you for your comments.
- We changed "lustra" to "luster" (lines 147 and 211).
- At the present it is difficult for us to estimate the quantum efficiency in 10 days. We will check the quantum efficiency in the near feature.
Reviewer 2 Report
A π-conjugated polymer based on CPDT and PCPDT-CO2H was reported in this manuscript. Hydroxyl radicals were generated in water by exposed the polymer to light irradiation, and exhibited certain photocatalytic property in the oxidation of organic chemicals. Both of them suggested that the prepared polymer was a potential candidate for metal-free organic heterogeneous photocatalyst. However, the present manuscript didn’t reach the requirement of publication and need to be improved.
1. Many works had been reported to construct organic light-sensitive materials including porphyrin, phthalocyanine and other compounds, and used in various fields. More introduction should be involved in “Introduction” to provide sufficient background to authors. Moreover, the aim of the present paper and the creativity compared with relative works was not clear, so it need to reorganize the “introduction” to improve them.
2. Energy band structure should be provided for the prepared polymer in addition to the UV-vis spectrum of the solid sample.
3. are there other active species during the irradiation in addition to hydroxyl radicals? The active species produced in light irradiation should be analyzed in detail and direct evidence including the EPR data should be provided.
4. Only FTIR was used to characterize the prepared polymer. More characterization should be adopted to demonstrate the successful synthesis of the PCPDT-CO2H polymer.
5. 1H NMR isn’t enough to demonstrate the photocatalytic products of arylboronic acid. Chromatography such as HPLC, GC or coupled with MS should be adopted to evaluate the products.
6. More evidences should be provided to support the proposed mechanism for the hydroxylation of phenylboronic acid (Figure 9).
7. Energy band structure should be provided for the prepared polymer.
Author Response
Thank you for your comments.
- We revised the introduction section according to reviewer's suggestion to clarify the aim of this research (from line 20 to line 86).
- We added optical bandgap determined from UV-vis-NIR absorption spectrum (line 220).
- As the reviewer pointed out it is important to examine the reaction system by EPR. However, EPR is not available in our research group at the present.
- Since the solubility of PCPDT-CO2H was not good, NMR characterization was unsuccessful. However, UV-vis-NIR spectrum strongly suggested the formation of π-conjugated structure.
- We checked published papers which examined photocatalytic hydroxylation of arylboronic acids to find that NMR analysis was enough for characterization of the reaction products.
- We are now undergoing further experiments to reveal the reaction mechanism.
Reviewer 3 Report
The manuscript reports a new metal free heterogeneous photocatalyst based on pí –conjugated polymer. New polycyclopentadithiophene derivative was prepared and tested in oxidative hydroxylation of arylboronic acid and in oxidation of benzaldehyde as model reactions. The manuscript is well-written and scientifically sound, with data supported conclusions. I recommend its publication.
I have a few comments only:
1. The polymer (in)solubility in DMF should be notice (DMF is a solvent for photocatalytic reaction.
2. What does it mean the sharp peak in the UV-vis spectrum (Fig.6)? Some impurity?
3. The spectrum in Fig.8 was used for determination of product composition. What signals were used for it exactly? Taking into account signals f and h, product 3 seems to be a minor product (in contradiction to Table 1). Maybe an integrated spectrum will be more informative.
Author Response
Thank you for your comments.
- We added the description of solubility in DMF (line 216).
- We added the description regarding the sharp peak in Figure 6 (from line 218 to line220).
- We revised the Table 1 and we added the description about the determination of the reaction products ratio.
Round 2
Reviewer 2 Report
It can be accepted for publication in Polymers.